# OpenReview forum: "Program of Thoughts Prompting: Disentangling Computation from Reasoning for Numerical Reasoning Tasks"
_TMLR — Accepted by TMLR_

### Review · Reviewer_Dt1q · 2023-07-13

**Summary Of Contributions:**

This work proposes a program of thoughts (PoT) technique, which uses large language models such as GPT-3.5 and Codex to generate text and programming language statements before inferring the answer. The programming language statements are executable by the Python interpreter. PoT is evaluated on five math word problem datasets and three financial-QA datasets in both few-shot and zero-shot settings. Results show that PoT achieves substantial performance gains over previous methods, and PoT has been further enhanced by self-consistency decoding.

This work advances the research in the field of mathematical reasoning by proposing a technique called PoT, which induces large language models to generate programs to express complex reasoning procedures before producing the answer. Compared with previous chain-of-thought techniques, PoT allows computation to be separately handled by an external program interpreter. Therefore, it is more suitable for arithmetic problems that require accurate calculations. This work also reports impressive results on widely-used arithmetic datasets.

Beyond the performance gains, this paper found that associating semantic meaning to the variable names is critical, which performs much better than directly using variable names.  In addition, breaking down the target equation into multiple reasoning steps helps boost performance, as prompting LLMs to directly generate the target equations is very challenging. The error analysis reveals that PoT still suffers from value grounding errors and logic generation errors, which indicate prospect optimization directions for future studies.

**Audience:**

Yes

**Broader Impact Concerns:**

I do not find any broader impact concerns.

**Claims And Evidence:**

Yes

**Requested Changes:**

1. Comparing existing studies that also prompt LLMs to generate programs to solve reasoning problems.

2. Resolving the confusing and problematic claims.

3. Improving the presentation quality of the paper.

**Strengths And Weaknesses:**

**Strengths**

- Given the considerable attention of chain of thought prompting techniques in the community, this paper provides a reasonable alternative by wrapping natural language problems into programs and employing external tools to execute those programs. There are various benefits after using PoT. On one hand, the input could be simplified; thus saving the overhead against the length limit. On the other hand, PoT has the potential to correct mistakes in the intermediate steps before producing the answer.

- The paper conducts extensive experiments on five math word problem datasets and three financial datasets. These datasets cover various input formats, including text, tables, and conversation. This paper also takes into account the influence of different backend LLMs and learning paradigms (few-shot learning and zero-shot learning), and sensitivity to demonstration exemplars, which improves the credibility of the findings.

- This work reports impressive results compared with the standard CoT techniques for numerical reasoning tasks. The effectiveness of the proposed technique is well justified.

**Weaknesses**

- The proposed PoT technique --- while reasonable --- is based on the assumption that the concerned problem is decomposable into programs. It remains unclear what will happen if the assumption is not satisfied. Adding a clarification would help position the work with more clear and precise application scenarios.

- As there are many recent studies in the literature that have prompted LLMs by program synthesis or symbolization to solve reasoning problems [1-6], it is highly recommended to discuss those existing studies and show how this work differs from them either in the idea or methodology and compare the performance in the experiment table.

- There are still various questions, technical mistakes, and stylistic suggestions (see the detailed comments below) that are not resolved yet.

**Questions & Comments**

- This paper considers the Codex and text-davinci-002 as the backend LLMs, and Codex has achieved much better performance. As Codex has been officially shutting down, it is interesting to see if PoT also performs impressively with other public backends to help reproducibility, such as CodeGen [7].

- In Section 2.3: "For instance, in the left example in Figure 3, the program will generate a float number 2.05. However, adding 2.05 to 11 AM cannot be easily handled by a Python program." --- 2.05 is not shown in Figure 3. Please check if the text mismatches the figure.

- The implementation prompt is not unclear. In Figure 3, the prompt is "# Answer this question by implementing a solver() function \n def solver(): \n # Let's write a Python program step by step \n First we need to define the following variable", while in Section 3 the implemented prompt is claimed to be "Let's write a Python program step by step." The inconsistent prompts may raise confusion on reproducibility.

- "We generally write prompts for 10-20 examples, and then tune the exemplar selection on a small validation set to choose the best 4-8 shots for the full set evaluation" --- It is highly recommended to provide the selected exemplars to help reproduction.

- "Unlike Codex, GPT-3 is not optimized for generating programs, and one would expect degraded performance with GPT-3 as the LLM." --- The argument is incorrect.  GPT-3 represents text-davinci-002 in this work, which is a supervised instruction-tuned model based on code-davinci-002. It has already been optimized for generating programs.

**Suggestions**

- The presentation of this paper can be improved. This paper confuses the use of parenthetical citations and textual citations. This issue happens throughout the whole paper. Please proofread the paper to avoid those mistakes.

- When introducing the chain of thought prompting work, it is suggested to cite [8] as it introduced the basic idea before being rebranded "chain of thought."

**References**

[1] Gao, Luyu, Aman Madaan, Shuyan Zhou, Uri Alon, Pengfei Liu, Yiming Yang, Jamie Callan, and Graham Neubig. "Pal: Program-aided language models." In International Conference on Machine Learning, pp. 10764-10799. PMLR, 2023.

[2] Yao, Shunyu, Dian Yu, Jeffrey Zhao, Izhak Shafran, Thomas L. Griffiths, Yuan Cao, and Karthik Narasimhan. "Tree of thoughts: Deliberate problem solving with large language models." arXiv preprint arXiv:2305.10601 (2023).

[3] Cheng, Zhoujun, Tianbao Xie, Peng Shi, Chengzu Li, Rahul Nadkarni, Yushi Hu, Caiming Xiong et al. "Binding Language Models in Symbolic Languages." In The Eleventh International Conference on Learning Representations. 2022.

[4] Gou, Zhibin, Zhihong Shao, Yeyun Gong, Yelong Shen, Yujiu Yang, Nan Duan, and Weizhu Chen. "Critic: Large language models can self-correct with tool-interactive critiquing." arXiv preprint arXiv:2305.11738 (2023).

[5] Wang, Lei, Wanyu Xu, Yihuai Lan, Zhiqiang Hu, Yunshi Lan, Roy Ka-Wei Lee, and Ee-Peng Lim. "Plan-and-solve prompting: Improving zero-shot chain-of-thought reasoning by large language models." arXiv preprint arXiv:2305.04091 (2023).

[6] Yao, Shunyu, Jeffrey Zhao, Dian Yu, Nan Du, Izhak Shafran, Karthik R. Narasimhan, and Yuan Cao. "ReAct: Synergizing Reasoning and Acting in Language Models." In The Eleventh International Conference on Learning Representations. 2022.

[7] Nijkamp, Erik, Bo Pang, Hiroaki Hayashi, Lifu Tu, Huan Wang, Yingbo Zhou, Silvio Savarese, and Caiming Xiong. "CodeGen: An Open Large Language Model for Code with Multi-Turn Program Synthesis." In The Eleventh International Conference on Learning Representations. 2022.

[8] Nye, Maxwell, Anders Johan Andreassen, Guy Gur-Ari, Henryk Michalewski, Jacob Austin, David Bieber, David Dohan et al. "Show your work: Scratchpads for intermediate computation with language models." arXiv preprint arXiv:2112.00114 (2021).

---

> ### Author Response · Authors · 2023-07-15
> **Clarification**
>
> Thanks a lot for the prompt review. It's very constructive. However, there are a few points I want to clarify a bit:
> The initial draft of our paper came out in Nov 2022, which is contemporary or earlier than most of the mentioned related work [1-6]. We resubmitted our paper to TMLR, which causes the time gap. Actually, [1,4,5] all cited our work and [4,5] are actually follow-up of our work and used PoT as their baseline. We plan to add a separate table to compare with [1]'s results. But for [4, 5], they already have PoT as the baseline in their "Table 2". I'm not entirely sure how to deal with these scenario. It would be great if you could provide some suggestion.

---

> > ### Comment · Reviewer_Dt1q · 2023-07-15
> > **Response to the clarification about contemporary studies**
> >
> > Thanks for the quick clarification. It makes sense to compare those studies either by experiments or discussions in the related work. The proposed solution appears to be adequate.  As a rule of thumb, including such comparisons can enhance the overall systematic and comprehensive nature of the work, allowing readers to gain a better understanding of the field's rapid development --- which is so rapid that various contemporary studies emerge recently. The clarification will be taken into account during the recommendation process. At the same time, I would kindly request the authors to address the remaining technical and presentation concerns.

---

> ### Author Response · Authors · 2023-07-17
> **Author Response**
>
> Thanks for the constructive review again.
>
> 1. We have added the comparison with PaL in Table 5.
> 2. discuss the contemporary work in the related work.
> 3. We add results from different backend models like GPT-3.5-turbo and CodeGen.
> 4. We found that CodeGen's performance is significantly behind GPT family models.
> 5. We fix all the citation issues.
> 6. We change the "chain of thought" citation to "show your work" in the introduction.
> 7. We fix the other mentioned writing issues in your "Comment".

---

> > ### Comment · Reviewer_Dt1q · 2023-07-18
> > **Thanks for the response**
> >
> > Thanks for the edits which have addressed part of my concerns. As there are still some questions or concerns unresolved, I am looking forward to the detailed author response and the other reviews.

---

> > > ### Author Response · Authors · 2023-07-19
> > > **Detailed response**
> > >
> > > Thanks for the quick response. Here are our detailed responses to your questions/comments.
> > >
> > > 1. "This paper considers the Codex and text-davinci-002 as the backend LLMs, and Codex has achieved much better performance. As Codex has been officially shutting down, it is interesting to see if PoT also performs impressively with other public backends to help reproducibility, such as CodeGen [7]."
> > >
> > > We have added more results from GPT-3.5-turbo and CodeGen-mono/multi into the paper. GPT-3.5-turbo gives reasonable improvement over Codex. However, CodeGen's results are still much worse than GPT-*.
> > >
> > > 2. In Section 2.3: "For instance, in the left example in Figure 3, the program will generate a float number 2.05. However, adding 2.05 to 11 AM cannot be easily handled by a Python program." --- 2.05 is not shown in Figure 3. Please check if the text mismatches the figure.
> > >
> > > Sorry for the confusion, 2.05 hour is computed from the program, which is not depicted in Figure 3. We add more sentences in Section 2.3 to explain this part.
> > >
> > > 3. The implementation prompt is not unclear. In Figure 3, the prompt is "# Answer this question by implementing a solver() function \n def solver(): \n # Let's write a Python program step by step \n First we need to define the following variable", while in Section 3 the implemented prompt is claimed to be "Let's write a Python program step by step." The inconsistent prompts may raise confusion on reproducibility.
> > >
> > > We have adjusted the Section 3 to be consistent with Figure 3. And Figure 3 is the final prompt we used.
> > >
> > > 4. "We generally write prompts for 10-20 examples, and then tune the exemplar selection on a small validation set to choose the best 4-8 shots for the full set evaluation" --- It is highly recommended to provide the selected exemplars to help reproduction.
> > >
> > > The best prompt is already added to our Github. We could also add the candidates into Github later.
> > >
> > > 5. "Unlike Codex, GPT-3 is not optimized for generating programs, and one would expect degraded performance with GPT-3 as the LLM." --- The argument is incorrect. GPT-3 represents text-davinci-002 in this work, which is a supervised instruction-tuned model based on code-davinci-002. It has already been optimized for generating programs.
> > >
> > > We have removed this paragraph. We are aware that text-davinci-002 is instruction-tuned on top of code-davinci-002. It seems that the instruction-tuned will hurt the model's capabilities to perform in-context code generation.
> > >
> > > 6. The proposed PoT technique --- while reasonable --- is based on the assumption that the concerned problem is decomposable into programs. It remains unclear what will happen if the assumption is not satisfied. Adding a clarification would help position the work with more clear and precise application scenarios.
> > >
> > > Thanks for the constructive feedback. Yes, we have the assumption that the problem can be solved at least partially by program. For the problem in AQuA QA, the question requires the model to first do calculation with programs, and then condition on the executed results to perform chain of thoughts prompting further. We have added some clarification in the discussion section.
> > >
> > > 7. As there are many recent studies in the literature that have prompted LLMs by program synthesis or symbolization to solve reasoning problems [1-6], it is highly recommended to discuss those existing studies and show how this work differs from them either in the idea or methodology and compare the performance in the experiment table.
> > >
> > > Thanks for pointing out these references. We compare with PaL in the experimental section. We also discuss the relation between our work and other contemporary work.

---

> > > > ### Comment · Reviewer_Dt1q · 2023-08-22
> > > > **Thanks for the response**
> > > >
> > > > Thank you for the detailed response. My concerns have been addressed.

---

### Review · Reviewer_k1bc · 2023-07-20

**Summary Of Contributions:**

This paper introduces program-of-thought prompting in which a large language model (LLM), instead of generating rationales in natural language, generates programs that can be executed by an external program interpreter. The paper argues that LLMs are error prone when performing numerical computation (esp with inputs of large magnitude) and hence it proposes to decouple computation from reasoning. The authors rigorously evaluate their method on 8 datasets and find that their performance outperforms Chain-of-thought (CoT) based prompting strategies significantly. The LLM used for the model is Codex ((code-davinci-002). They also compare to  GPT-3 (text-davinci-002), ChatGPT (gpt-turbo-3.5), and CodeGen.

**Audience:**

Yes

**Broader Impact Concerns:**

There is a limitation section which I thought was well-written.

**Claims And Evidence:**

Yes

**Requested Changes:**

In sec 2.3, it is unclear how you make the model decide when to stop or use the intermediate result and the question to continue prompting. Also, in the specific example used, it seems like no additional prompting is needed because in Fig 3 (left), it seems the model is capable of automatically transforming 7:30 to 7.5, therefore it should also be able to do the reverse i.e. convert 2:05 to HH:MM. Please address this in the next draft as knowing when to stop or continue prompting is very critical to understand how your model operates and will help me make the decision for the paper.
Page 5 (Therefore, we suppress the ‘#’ token logits by -2…) -> can you mention this before in the paper (e.g. in Sec 2.2). This is an important detail and an average reader might miss it. Can you also justify how you came up with ”-2”? Was it done using a validation set?
Sec 3.2 - How do you explain the difference in performance between CoT + calc and PoT? Is it because PoT is explicitly prompted to generate programs? Did you try using the same prompts for CoT and PoT experiments?


**Strengths And Weaknesses:**

Strengths:
A reasonable extension of chain-of-thoughts prompting strategy that makes a lot of sense (letting LLMs generate executable statements that are executed by external tools and not the LLMs)
Good performance improvance wrt CoT on a variety of datasets where the input modalities vary from text to tables and conversations and covers two domains - math word problems and finance.
The paper is very clear and easy to understand
Weakness:
It would be nice to differentiate this paper’s contribution from LLMs ability to use external tools (e.g. APIs). Seen through that lens, this paper investigates the ability of LLMs to produce inputs (programs) for a particular kind of tool (python interpreter). Is that a correct way to see it? If not, how do you differentiate this work?

---

> ### Author Response · Authors · 2023-07-21
> **Author Response**
>
> Thanks a lot for the detailed review
>
> 1. "It would be nice to differentiate this paper’s contribution from LLMs ability to use external tools (e.g. APIs). Seen through that lens, this paper investigates the ability of LLMs to produce inputs (programs) for a particular kind of tool (python interpreter). Is that a correct way to see it? If not, how do you differentiate this work?"
> Thanks for bringing up this discussion. As we replied to another reviewer, our initial paper came out in Nov 2022. This TMLR submission is a re-submission, which causes the time gap. The famous Toolformer and ART paper came out in Feb/March 2023, which are actually successor of our PoT paper. ART (https://arxiv.org/abs/2303.09014) actually cites us and uses our released code extensively. Our paper is amongst the earliest work to use tools to solve domain-specific problems.
>
> 2. "In sec 2.3, it is unclear how you make the model decide when to stop or use the intermediate result and the question to continue prompting  ...":
> Sorry for the confusion. This is actually very simple and intuitive. We mainly use this in AQuA QA dataset because the other datasets don't even need this. For AQuA QA, Codex learns when to stop or require additional CoT through the demonstrations. In the demonstration, I will have example containing both "PoT program. **This is the answer**", "PoT program. **Please continue to **prompt**". In the prompting process, we will detect whether the model is outputting this marker to decide next step. If we see **This is the answer**, we will compare the answer with all the answer options (string match) in AQuA to make prediction. If we see **Please continue to **prompt**, we will first execute the program output and then feed it to do CoT to choose the best answer option.
>
> 3. "it seems the model is capable of automatically transforming 7:30 to 7.5, therefore it should also be able to do the reverse i.e. convert 2:05 to HH:MM. ".
> Yes, conceptually, we can write a program to convert floating point number into HH:MM format, but we believe such format changing can be better handled by CoT. Our motivation to get the best out of PoT and CoT. We breakdown the question into computation part and text reasoning part. The PoT will first be called to handle the computation part. If  **Please continue to **prompt** is detected, we will continue to use CoT to carry out the rest text reasoning part. Our early experiment shows that such a breakdown leads to best performance.
>
> 4. "Page 5 (Therefore, we suppress the ‘#’ token logits by -2…) -> can you mention this before in the paper (e.g. in Sec 2.2). This is an important detail and an average reader might miss it. ":
> Sure, we will do it.
>
> 5. Can you also justify how you came up with ”-2”?
> Yes, we use a few examples from the training set to adjust this hyper-parameter.
>
> 6. Sec 3.2 - How do you explain the difference in performance between CoT + calc and PoT?
> "CoT + calc" is a through regular expression to do post-processing. It might be able to correct CoT output "Alice has 20 apples and 33 pears, therefore she has 20+33=63 fruits" by finding the "*=*", and then use calculator to fix "63" into "53". However, if the CoT output is "Alice has 20 apples and 33 pears, therefore she has 63 fruits", the regular expression won't detect where the addition happens, thus not able to fix it through post-processing.
> In contrast, PoT already format everything into program to ensure everything is being executed accurately.
>
> 7. Did you try using the same prompts for CoT and PoT experiments?
> I'm not entirely sure about "same prompt". Do you mean using the same "demonstration question"? But the reasoning chain are in different formats (language vs. program)? If that's what you meant, the current experiments are already doing this.

---

> > ### Comment · Reviewer_k1bc · 2023-07-25
> > **Thanks for the response**
> >
> > Thank you for the detailed response.
> >
> > 1. I see, I wasn't aware of the exact dates - this makes sense. My recommendation would be to cite them as contemporary work and mention that this is a specific case of tool usage with LLMs. I dont think that reduces the impact of any claims made by the paper.
> >
> > 2. Got it, thank you. Can you add this detail into the paper (either in the main section or the appendix whichever makes sense). This actually cleared up a lot of the following questions I asked so I think this might help other readers too.
> >
> > 3. Makes sense as well, thank you
> >
> > 5. Please mention this detail in the paper about how you came up with -2.
> >
> > 6. Totally makes sense after this explanation. I wonder if this can be added to the paper as well. This also clears up my question regarding 7. above.

---

> > > ### Author Response · Authors · 2023-07-25
> > > **Made the changes**
> > >
> > > Thanks a lot for the constructive feedback.
> > >
> > > 1. I have added the discussion in the related section.
> > > 2. Yes, we have expanded that section to include more details about this case.
> > > 4. I have added a sentence to explain why we chose -2.
> > > 5. I have added a few sentence to explain why "CoT+calculator" is not ideal solution.

---

> > > > ### Comment · Reviewer_k1bc · 2023-08-23
> > > > **Thank you**
> > > >
> > > > Thank you, all my concerns have been addressed at this time.

---

### Review · Reviewer_7mvy · 2023-08-21

**Summary Of Contributions:**

The paper introduces an innovative approach to LLM prompting, termed PoT, which harnesses the code generation capabilities of LLMs to address mathematical challenges. The study showcases their experimental work on PoT, contrasting the outcomes with the conventional CoT methodology, revealing substantial enhancements in performance.

**Audience:**

Yes

**Broader Impact Concerns:**

I have no ethical concerns in this work.

**Claims And Evidence:**

Yes

**Requested Changes:**

1. Describe more on how the "#" is suppressed in the research.
2. Provide illustrative examples showcasing instances of few-shot learning exemplars.

**Strengths And Weaknesses:**

Strengths:
1). This paper introduces a novel method for assessing LLMs, utilizing their coding proficiencies to tackle mathematical problems. Extensive comparative analysis between PoT and CoT underscores the potency of PoT through multiple dimensions of evaluation.
2). The investigation into PoT is thorough, encompassing an in-depth comparation between PoT and CoT. Additionally, it explores the prowess of PoT in few-shot learning scenarios involving varying shot quantities.

Weaknesses:
1). Certain technical intricacies require more lucid elucidation. For instance:
       a) Clarification is needed regarding the term "bias" in the context of "-2 as the bias" mentioned on page #6, first line.
       b) Illustrative instances of "few-shot" applications could enhance comprehension.
2). The paper's impact could be strengthened if a similar study of GPT-4 is conducted with it's exterior applications.

---

> ### Author Response · Authors · 2023-08-22
> **Author Response**
>
> Thanks a lot for providing insightful reviews.
>
> - a) Clarification is needed regarding the term "bias" in the context of "-2 as the bias" mentioned on page #6, first line.
>
> This is by simply setting `logit_bias' argument in the `openai.Completion.create' function (the # vocab index is 1303, we set the argument to -2 to realize the suppression).
>
> - b) Illustrative instances of "few-shot" applications could enhance comprehension.
>
> Do you mean putting the "few-shot" demonstrations in the paper? It's actually pretty long. It's already been open-sourced in our Github repo. It should be relatively easy for people to visualize using our provided script. Of course, we could select some to put in our Appendix.
>
> - c)  The paper's impact could be strengthened if a similar study of GPT-4 is conducted with its exterior applications.
>
> Thanks a lot for the reminder. We have already conducted GPT-4 experiments with CoT on the GSM, AQuA and FinQA dataset. We show our results as follows:
>
> | Method         | GSM    | AQuA  | FinQA |
> |----------------|---------|---------|---------|
> | GPT4 (CoT)  |  92.0   |   72.4    |  58.2   |
> | GPT4 (PoT)  |  97.2   |   84.4    |  74.0   |
>
> As can be seen, PoT is also very effective for GPT-4, the improvement on all these datasets are very significant.

---

> > ### Comment · Reviewer_7mvy · 2023-08-22
> >
> > a) Thank you for the clarification, could you briefly illustrate how this -2 is chosen?
> > b) It would be great if you could select one or two examples in the appendix.
> > c) The result is very impressive, could you include this in the paper?

---

> > > ### Author Response · Authors · 2023-08-22
> > > **Response**
> > >
> > > Hi there,
> > >
> > > a) Thank you for the clarification, could you briefly illustrate how this -2 is chosen?
> > >
> > > We didn't tune this much. We set the bias to -1, -2, -3, -5. And then we observe the outputs on held-out 10 examples to see how much of them are actually executable code. Through this, we found -2 is roughly the best.
> > >
> > > b) It would be great if you could select one or two examples in the appendix
> > > Sure, no problem.
> > >
> > > c) The result is very impressive, could you include this in the paper?
> > > Of course, we will update the pdf accordingly.

---

> > > > ### Author Response · Authors · 2023-08-25
> > > > **Updated the new version**
> > > >
> > > > Thanks! We have updated the pdf to include the GPT-4 results and the exemplars in the Appendix.

---

### Decision · Action_Editors · 2023-10-15

**Recommendation:** Accept as is

**Comment:**

The paper provides a clear contribution, backed up by strong empirical evidence. The authors have addressed all the reviewer comments in their response and revised version. The reviewers unanimously recommend acceptance.

**Audience:**

This paper will substantially inform future work into the use of LLMs to solve numerical reasoning tasks, as well as for program induction in general, which will be of broad interest to the TMLR audience.

**Claims And Evidence:**

This paper proposes Program of Thoughts (PoT), a technique modeled on the popular Chain-of-thought paradigm for using large language models (LLMs) to reason while solving problems. The key innovation here is getting LLMs to generate intermediate programs, which can be executed (say using a Python interpreter) to produce accurate intermediate answers, something that can be tricky for LLMs to generate, especially for math problems. The experiments are convincing and convincingly demonstrate the empirical usefulness of this techniques.